# Integration of Metabolomics and Transcriptomics to Explore Dynamic Alterations in Fruit Color and Quality in ‘Comte de Paris’ Pineapples during Ripening Processes

**DOI:** 10.3390/ijms242216384

**Published:** 2023-11-16

**Authors:** Kanghua Song, Xiumei Zhang, Jiameng Liu, Quansheng Yao, Yixing Li, Xiaowan Hou, Shenghui Liu, Xunxia Qiu, Yue Yang, Li Chen, Keqian Hong, Lijing Lin

**Affiliations:** 1Key Laboratory for Postharvest Physiology and Technology of Tropical Horticultural Products of Hainan Province, South Subtropical Crop Research Institute, Chinese Academy of Tropical Agricultural Sciences, Zhanjiang 524091, China; sun_nykang@163.com (K.S.); asiazhang1975@163.com (X.Z.); yqsh1028@163.com (Q.Y.); yimiyangguanghxw@163.com (X.H.); 13652860494@126.com (S.L.); yangyue0303@163.com (Y.Y.); chenroumu@126.com (L.C.); 2Hainan Key Laboratory of Storage & Processing of Fruits and Vegetables, Agricultural Products Processing Research Institute, Chinese Academy of Tropical Agricultural Sciences, Zhanjiang 524001, China; jmengliu@163.com (J.L.); qxx4503@163.com (X.Q.); 3Institute of Tropical Bioscience and Biotechnology, Chinese Academy of Tropical Agricultural Sciences, Haikou 571101, China; yixing0221@163.com

**Keywords:** pineapple, ripening, yellowing, fruit quality, metabolomics, transcriptomics

## Abstract

Pineapple color yellowing and quality promotion gradually manifest as pineapple fruit ripening progresses. To understand the molecular mechanism underlying yellowing in pineapples during ripening, coupled with alterations in fruit quality, comprehensive metabolome and transcriptome investigations were carried out. These investigations were conducted using pulp samples collected at three distinct stages of maturity: young fruit (YF), mature fruit (MF), and fully mature fruit (FMF). This study revealed a noteworthy increase in the levels of total phenols and flavones, coupled with a concurrent decline in lignin and total acid contents as the fruit transitioned from YF to FMF. Furthermore, the analysis yielded 167 differentially accumulated metabolites (DAMs) and 2194 differentially expressed genes (DEGs). Integration analysis based on DAMs and DEGs revealed that the biosynthesis of plant secondary metabolites, particularly the flavonol, flavonoid, and phenypropanoid pathways, plays a pivotal role in fruit yellowing. Additionally, RNA-seq analysis showed that structural genes, such as *FLS*, *FNS*, *F3H*, *DFR*, *ANR*, and *GST*, in the flavonoid biosynthetic pathway were upregulated, whereas the *COMT*, *CCR*, and *CAD* genes involved in lignin metabolism were downregulated as fruit ripening progressed. *APX* as well as *PPO,* and *ACO* genes related to the organic acid accumulations were upregulated and downregulated, respectively. Importantly, a comprehensive regulatory network encompassing genes that contribute to the metabolism of flavones, flavonols, lignin, and organic acids was proposed. This network sheds light on the intricate processes that underlie fruit yellowing and quality alterations. These findings enhance our understanding of the regulatory pathways governing pineapple ripening and offer valuable scientific insight into the molecular breeding of pineapples.

## 1. Introduction

Pineapple (*Ananas comosus* L. Merr.), a member of the Bromeliaceae family, ranks as the third most commercially significant tropical and subtropical fruit globally, earning widespread acclaim for its nutritional value [1,2]. Biologically categorized as a non-climacteric species, pineapple does not undergo a post-ripening process and, post-harvest, progressively experiences a decline in quality acceptability. However, pineapple fruit ripening is accompanied by a gradual increase in respiration and ethylene release, resembling the behavior of climacteric fruits that undergo transformations in appearance, texture, and nutrient metabolism [3,4].

As aggregate fruit with a unique ripening profile, understanding the ripening characteristics of pineapple is important from both scientific and market perspectives. The transition in pulp color from white to yellow is believed to be a complex interplay between carotenoids and flavonoids, flavonoids being the predominant pigments [5]. The texture of fleshy fruit, a pivotal quality attribute, not only constitutes a vital component of the primary cell wall polysaccharide but also significantly contributes to fruit hardness and overall edibility [6]. Organic acids, which comprise diverse components, play a pivotal role in regulating fruit flavor. While some aspects of the content and alteration patterns of organic acids in pineapple fruit have been investigated in detail [7], questions remain regarding which components change as the fruit matures.

Furthermore, the biosynthesis of phenylpropanoid and phenylpropanoid, along with organic acid metabolism, coincidentally, three crucial pathways responsive to oxidative stress, result in subsequent alterations in color, texture, and flavor, driven by the production of reactive oxygen species (ROS) during pineapple fruit ripening [8,9]. For instance, redox mechanisms have been identified as critical in the ripening of the Smooth Cayenne variety [10]. Antioxidant compounds, such as flavonoids and polyphenols, have also been extensively investigated in various pineapple cultivars, including Smooth Cayenne, Red Spanish, MD-2, and Queen, during fruit ripening [11,12,13,14], underscoring the association between ROS derived from plant secondary metabolites and diverse ripening responses.

Despite prior investigations into alterations in cellular lipids, including lipid degradation and peroxidation, during fruit ripening in the ‘Comte de Paris’ cultivar, the chief pineapple variety cultivated in China, little is known about the molecular events and metabolites involved in transformations associated with fruit yellowing, softening, and quality improvement during ripening [15]. Variations in pigments, especially flavonoids and total phenols, attributed to fruit yellowing have not been comprehensively investigated and need further elucidation. Therefore, the primary objective of this study was to examine ripening-related physicochemical parameters, including color, phenols, flavones, lignin, and quality. These findings were subsequently integrated with metabolomics and transcriptomics to elucidate the corresponding metabolites and genetic factors that govern fruit ripening. This comprehensive investigation is expected to significantly advance our understanding of the alterations in fruit color and quality during the fruit ripening process in ‘Comte de Paris’ pineapples, ultimately offering valuable insights into the molecular breeding of this fruit.

## 2. Results

### 2.1. L*, a*, and b* Values of Peel and Pulp

As illustrated in Figure 1A, noticeable alterations in the morphology of both the peel and pulp color were observed during the transition from green to yellow, and white to yellow, respectively, in the fruit’s progression from the YF stage to the FMF stage. Notably, the L* value of the peel exhibited a prominent increase, whereas the L* value of the pulp exhibited a significant decrease. These differences in L* values were statistically significant as the fruit transitioned from YF to FMF, in both the peel and pulp.

Additionally, the values of a* and b* in both the peel and pulp exhibited similar trends, consistently and significantly increasing as the ripening process advanced, as depicted in Figure 1B(a–f).

### 2.2. Contents of SSC, TA, and Vc in Pulp

The TSS content exhibited a noteworthy increase from the YF to the MF stage, and a significant difference was observed between these two stages. However, as the fruit progressed from the MF to FMF stage, there was no substantial change in the TSS content (Figure 1B(g)).

In contrast, both TA and Vc contents displayed a marked decrease in the MF compared with the YF, and subsequently, their levels remained relatively stable as the fruit matured further (Figure 1B(h,i)).

### 2.3. Contents of Total Phenolic, Total Flavonoids, and Lignin in Pulp

The contents of total phenol and flavonoids displayed a consistent trend, gradually increasing as the fruit matured. Significant differences were observed between the YF and MF stages, and between the MF and FMF stages. Consequently, their contents reached higher levels in the FMF (Figure 1B(j,k)).

Conversely, the accumulation level of lignin exhibited a marked decrease in the YF compared with the MF. Subsequently, there was a slight increase, with significant differences detected between the MF and FMF (Figure 1B(l)).

### 2.4. Differentially Accumulated Metabolites (DAMs) Annotated

To gain deeper insights into the metabolites associated with yellow pigment formation, softening, and acidity decrease during the ripening process, we re-evaluated the metabolomics data using YF, MF, and FMF as materials [15]. By conducting a comprehensive comparison of the metabolites in the pulps of these three stages, a total of 167 DAMs were identified and categorized into various groups. These included 31 lipids, 28 organic acids, 27 phenylpropanoids, 21 flavonoids, 19 amino acids and derivatives, 10 nucleotides and derivatives, 10 alkaloids, 5 saccharides and alcohols, 4 vitamins, 3 tannins, and 9 other unclassified substances (Figure 2A).

Furthermore, we quantified the numbers of upregulated and downregulated DAMs when comparing YF with MF, YF with FMF, and MF with FMF. Subsequently, there were 61 upregulated and 52 downregulated DAMs between the YF and MF stages, 66 upregulated and 51 downregulated DAMs between the YF and FMF stages, and 58 upregulated and 26 downregulated DAMs between the MF and FMF stages (Figure 2B).

### 2.5. DAMs Analysis

To further elucidate the characteristics of the 167 DAMs, we conducted a *K*-means clustering analysis, which resulted in the creation of six clusters designated as C1–C6. Notably, clusters C1, C2, and C4 generally exhibited downward trends, whereas clusters C3, C5, and C6 showed an upward trend during fruit ripening (Figure 3A).

We further analyzed the distribution of metabolites from 11 categories within C1–C6. Specifically, within the clusters showing an upward trend during fruit ripening (C3, C5, and C6), 13 flavonoids, 20 organic acids, 16 phenylpropanoids, and 2 vitamins displayed increasing levels as the fruit matured. Conversely, in the clusters exhibiting a downward trend (C1, C2, and C4), the levels of 8 flavonoids, 8 organic acids, 11 phenylpropanoids, and 2 vitamins decreased (Figure 3B).

Furthermore, we conducted a Kyoto Encyclopedia of Genes and Genomes (KEGG) enrichment analysis of 167 DAMs and performed clustering analysis to assess the changing trends of enriched pathways in the YF, MF, and FMF stages. This analysis revealed that pathways such as the biosynthesis of secondary metabolites, phenylpropanoid biosynthesis, flavone and flavonol biosynthesis, anthocyanin biosynthesis, and ascorbate and aldarate metabolism were significantly enriched (Figure 3C).

Taking a closer look at the DAMs involved in the biosynthesis of secondary metabolites, we observed that organic acids such as L-ascorbate, cis-aconitic acid, rosmarinic acid, sebacate, mandelic acid, suberic acid, and quinic acid O-di-glucuronic acid were degraded during the ripening of the pineapple fruit. In the DAMs related to flavonoid biosynthesis, metabolites such as kaempferol 3-O-glucoside, ayanin, apigenin 7-O-glucoside, and petunidin 3-O-glucoside, belonging to the flavone and flavonol categories, displayed increasing trends as the fruit ripened. Conversely, within the DAMs of phenylpropanoid biosynthesis, phenol compounds, such as cinnamic acid and caffeic acid, steadily accumulated as the fruit matured. Additionally, three lignin monomers, sinapyl alcohol, *ρ*-coumaryl alcohol, and coniferyl alcohol, were identified. Among these, *ρ*-coumaryl alcohol exhibited a dramatic accumulation from the MF to FMF stages, coniferyl alcohol reached its highest level in the FMF stage, and the sinapyl alcohol content continuously decreased during fruit ripening (Figure 3D).

### 2.6. Transcriptome Profiling

#### 2.6.1. Transcriptomics Analysis

To delve into the potential transcriptional modulatory mechanisms governing the processes of turning yellow and the alteration in fruit quality during pineapple ripening, a total of nine cDNA libraries were constructed from total RNAs for high-throughput RNA-seq analysis. After the removal of low-quality reads and adaptor sequences, the resulting total clean data ranged from 44,474,662 to 72,092,510 for each library, with a retention rate of 98.37% to 98.83%. The Q20 percentage reached 98.29%, and the GC content was recorded at 52.67%. Notably, for each library, approximately 62.00% to 75.17% of clean reads were successfully mapped onto the reference genome (Supplemental Appendix A).

Moreover, the principle component analysis (PCA) depicted in Figure 4A reveals all biological replicates together, demonstrating high consistency between the replicates. Importantly, substantial differences were observed between the YF and MF stages, YF and FMF stages, and MF and FMF stages, underscoring the transcriptional variations associated with fruit ripening.

To further assess the differences in transcriptome profiles between the three stages (YF, MF, and FMF), we conducted a hierarchical clustering analysis (HCA), which resulted in the formation of three distinct clusters (Figure 4B). These findings collectively affirm the reliability of our bioinformatics analysis of our RNA-seq data obtained from different stages of pineapple fruit ripening.

#### 2.6.2. DEGs Analysis

Applying a threshold of log2 |Fold Change| ≥1 and FDR < 0.05, the RNA-seq data analysis identified and annotated a total of 2194 DEGs across the three samples. The results, as presented in Figure 5A,B, illustrate the number of DEGs resulting from sequence comparisons between YF and MF, YF and FMF, and MF and FMF, with 989, 1796, and 707 DEGs, respectively. Notably, the largest number of DEGs was observed in the transition from YF to FMF, followed by YF to MF, and MF to FMF. This suggests that there is a greater involvement of genes in metabolic regulatory responses during the transition from MF to FMF than those of YF to MF, and MF to FMF.

Subsequently, we conducted a *K*-means analysis of the DEGs, resulting in their categorization into 10 clusters designated as T1–T10. Among these clusters, T1, T4, T6, T7, and T10 displayed upward trends in gene expression, while T2, T3, T5, T8, and T9 showed downward trends (Figure 5C).

Furthermore, we performed a KEGG enrichment analysis for DEGs across the 10 clusters, revealing a significant distribution of DEGs in pathways related to the biosynthesis of the secondary metabolites, phenylpropanoid biosynthesis, flavone and flavonol biosynthesis, and ascorbate and aldarate metabolism (Figure 5D). These findings highlight the involvement of these pathways in the regulatory processes associated with pineapple fruit ripening.

### 2.7. WGCNA Analysis and Gene Network Visualization

To delve deeper into the genetic foundations governing the development of yellow pigment, softening, and acidity changes throughout the fruit ripening process, we employed WGCNA (version 1.69) to construct unsigned co-expression networks. By leveraging the identified 2194 DEGs obtained from the comparative analysis between the three stages of pineapple fruit maturity, we identified six distinct gene expression modules.

Notably, four of these modules, specifically, MEblue, MEyellow, MEturquoise, and MEgreen, exhibited significant correlations with the changes observed in ripening-related traits, including total flavonoids content, total phenolic content, lignin levels, Vc content, and TA (Figure 6A,B). These correlations suggest these gene sets play a pivotal role in governing the observed alterations in pineapple fruit ripening-related traits.

### 2.8. RT-qPCR Validation of DEGs

To validate the accuracy of the RNA-seq data, we conducted an RT-qPCR analysis of 14 structural genes involved in the biosynthesis of flavonoids and lignin. These genes include cinnamoyl-CoA reductase (*CCR*, Aco015695, Aco026870, and Aco005397), cinnamyl alcohol dehydrogenase (*CAD* and Aco001032), caffeic acid 3-O-Methyltranferase (*COMT*, Aco000946, and Aco000839), 4-coumaroyl CoA ligase (*4CL* and Aco019975), laccase (*LAC* and Aco015409), peroxidase (*POD*, Aco008456, and Aco002056), flavanone 3-hydroxylase (*F3H* and Aco027900), anthocyanidin reductase (*ANR* and Aco010710), dihydroflavonol 4-reductase (*DFR* and Aco006769), and glutathione S-transferase (*GST* and Aco001260).

RNA samples extracted from pineapple pulp tissues at three different ripening stages served as templates for the RT-qPCR analysis. Correlation analysis revealed that a significant correlation coefficient of 0.878 between the RT-qPCR and RNA-seq data was obtained (Figure 7). This strong correlation suggests that the RT-qPCR results aligned well with the trends in gene expression levels detected with RNA-seq, providing additional confidence in the accuracy of the RNA-seq data.

## 3. Discussion

In recent decades, the maturation of pineapple fruit has garnered significant attention because of its complex ripening characteristics, which vary by variety and have considerable nutritional and commercial value [10,12,13,14]. Understanding the ripening characteristics of the ‘Comte de Paris’ pineapple, the dominant cultivar in China, is crucial for regulating fruit quality and fostering the development of the local pineapple industry.

In our study, the changing patterns of a* and b* values in both the peel and the pulp closely mirrored the trends in total phenol and flavonoid contents, exhibiting that fruit yellowing is positively related to phenol and flavonoid accumulations during ripening. Conversely, the TA and lignin contents displayed similar trends, all significantly decreasing from the YF stage to the MF stage (Figure 1). These observations are similar to findings in other fruits during ripening, such as jujube, strawberries, papaya, and peaches [16,17,18,19]. These findings suggest that the accumulation of flavonoids may contribute to pigment transition, which may be linked to the reduction in the lignin content in cell walls, and organic acid degradation may explain the decrease in acidity during ripening.

To gain deeper insights into the metabolites associated with fruit ripening, we investigated variations in metabolites during pineapple fruit ripening. The pineapple fruit contains a variety of flavonoids that are closely linked to its color characteristics. The presence of pigments, especially flavonols and flavones, is believed to contribute to the white-yellow pigmentation of pineapple fruit [20,21]. In our study, we observed a significant increase in flavonols, such as ayanin and kaempferol 3-O-glucoside (astragalin), during fruit ripening, along with flavones, such as chrysoeriol 5-O-hexoside, chrysoeriol O-hexosyl-O-hexosyl-O-glucuronic acid, acacetin O-acetyl hexoside, apigenin 7-O-glucoside (cosmosiin), tricin O-malonylhexoside, tricin O-sinapoylhexoside, and nobiletin. Additionally, other flavanones, including naringenin and hesperidin, as well as anthocyanin such as petunidin 3-O-glucoside, all exhibited continuous accumulation as fruit ripening progressed (Figure 3 and Figure 8). These trends were consistent with the observed changes in fruit color and total flavonoid content. Similar results were observed for flavonoids, and anthocyanin presented an upward tendency in *Lycium chinense* fruit during development [22], while it has been reported that total flavonoid contents declined in *Prunus humilis* as fruit ripening progressed [23]. These findings suggest that flavonoid metabolism is an intricate process, and different fruits possess specific networks in governing the biosynthesis of flavonoids during ripening. Thus, further research is needed to identify the specific components that contribute the most to yellow pigmentation.

Fruit texture is closely correlated with lignin content in the secondary cell wall [24,25]. In this study, we identified three lignin monomers in DAMs: sinapyl alcohol, *p*-coumaryl alcohol, and coniferyl alcohol. Sinapyl alcohol exhibited a similar trend to lignin content, whereas *p*-coumaryl alcohol and coniferyl alcohol displayed the opposite trend. Concurrently, phenolic compounds, such as cinnamic acid and caffeic acid, derived from phenylpropanoid biosynthesis, generally increased as fruit ripening progressed, consistent with the change in the total phenolic content (Figure 3). These findings suggest that these compounds play crucial roles in lignin and phenol biosynthesis.

A dramatic drop in the organic acids content from YF to MF was observed in our study. Although the exact composition of the organic acids was not determined, previous research has indicated that citric acid is the predominant organic acid in mature pineapple fruit [26,27,28]. In our study, we identified cis-aconitic acid, which is an intermediate in the conversion of citric acid to isocitric acid in the tricarboxylic acid cycle, as it significantly changes with fruit ripening. Ascorbic acid, which participates in the oxidative processes during fruit ripening, also decreased as fruit ripening advanced [29]. Additionally, mandelic acid, rosmarinic acid, suberic acid, sebacate, and quinic acid O-di-glucuronic acid significantly decreased, suggesting that the reduction in TA may be attributed to the decreased contents of these organic acids during pineapple fruit ripening.

Through our metabolomics and transcriptomics analyses, key genes involved in the modulations of metabolites, comprising total flavonoids, total phenolic content, lignin, and organic acids, were characterized.

The biosynthesis of flavonoids has been well documented from a molecular genetic perspective [30,31]. Chalcone isomerase (CHI) catalyzes the conversion of colorless chalcones to flavanone (naringenin). In monocots, naringenin is further metabolized into flavones by flavone synthase (FNS), where F3H and flavonol synthase (FLS) are required for the synthesis of dihydroflavonols and flavonols [32]. In some plant tissues such as fruits, DFR competitively reduces dihydroflavonols to leucoanthocyanidins, which are then converted to proanthocyanidins by ANR [26]. GST is responsible for transferring anthocyanins, flavonols, and flavones into the vacuole or cell wall for storage [33,34]. In our study, nine oxidoreductase genes including *FLS* (Aco027900), *FNS* (Aco019006), *F3H* (Aco001560 and Aco006882), *DFR* (Aco006769), *ANR* (Aco010710), and *GST* (Aco001260, Aco013918, Aco005105, and Aco013915), were identified via WGCNA (Figure 6). This observation is similar to that reported by Luo et al. (2021), who showed that the *PsFLSs* and *PsF3Hs* genes exhibited upregulated expressions in tree peonies during the developmental process [35]. Combining these results, we infer that these structural genes exhibited significantly and continuously increasing expression during fruit ripening, correlating with the accumulation of flavones and flavonols, which contribute to pigment formation.

Phenylpropanoid metabolism is closely associated with lignin biosynthesis. In this pathway, 4-coumaroyl CoA ligase (4CL) catalyzes the formation of activated thioesters of hydroxycinnamic acids, which enter different branch pathways of phenylpropanoid metabolism [36]. Hydroxycinnamoyl CoA: shikimate hydroxycinnamoyl transferase (HCT) is a key metabolic entry point for the synthesis of essential lignin monomers, coniferyl and sinapyl alcohols, particularly in monocotyledonous plant [37]. COMT methylates caffeic acid to ferulic acid, whereas CCR and CAD convert CoA ester to alcohol during monolignol biosynthesis. These monolignols are exported to the cell wall and polymerized into lignin by POD or depolymerized by LAC [38,39]. In our study, four hydroxycinnamoyl transferase (HCT) genes were downregulated during fruit ripening, consistent with the accumulation of *p*-coumaryl alcohol and the reduction in lignin monomers, coniferyl, and sinapyl alcohols. The expressions of the screened *COMT* (Aco017593 and Aco018902), *CCR* (Aco015695), and *CAD* (Aco001032) genes generally showed a decreasing trend, in line with the accumulation of intermediate phenolic metabolites such as caffeic acid and caffeoyl aldehyde. Additionally, the identified *POD* genes (Aco008465, Aco021646, Aco015271, Aco002056, Aco004613, Aco029136, Aco003045, Aco026779, Aco001617, Aco013666, Aco021355, Aco004784, and Aco021127) and the *LAC* gene (Aco015409) were downregulated during fruit ripening (Figure 8), indicating their potential role in the depolymerization of lignin in the cell wall as fruit ripening progressed.

Ascorbic acid is known to be oxidized to dehydroascorbic acid during fruit ripening, a process catalyzed by ascorbic acid oxidase (APX) [10,29,40]. In our study, genes related to *APX* (Aco007028) and polyphenol oxidase (*PPO* and Aco014848) exhibited consistently increasing expression as fruit ripening progressed, suggesting their involvement in the oxidation of ascorbic acid. Cis-aconitic acid, an organic acid, is catalyzed by aconitase (ACO), which converts citric acid to isocitric acid [41]. In our study, the expression of *ACO* (Aco009034) gradually decreased (Figure 8), suggesting its potential role in the reduced accumulation of cis-aconitic acid during pineapple fruit ripening.

Based on these findings, we propose a schematic overview to elucidate the metabolites and corresponding genes involved in pineapple fruit ripening (Figure 8). Among this network, the transcript abundance of a series of putative candidate genes associated with the metabolism of flavonoids, lignin, and organic acids, such as *FLS*, *FNS*, *F3H*, *DFR*, *ANR*, *GST*, *COMT*, *CCR*, *CAD*, *PPO*, and *ACO*, was induced as pineapple fruits matured, and this effect likely contributed to the increasing phenols and flavonoids contents, decreasing lignin and total acid, thus causing fruit yellowing, coupled with quality alterations.

## 4. Materials and Methods

### 4.1. Plant Materials and Sampling

Pineapple (*Ananas comosus* cv. ‘Comte de Paris’) fruits, carefully selected for their absence of defects and mechanical damage, were procured from a commercial orchard in Xuwen county (20°34′ N; 110°17′ E), Zhanjiang city, Guangdong province, China, in September 2018. These fruits were swiftly transported to the laboratory within a two-hour timeframe. A total of 120 fruits were used in this experiment, meticulously chosen to represent three distinct stages of maturity: young fruit (YF), mature fruit (MF), and fully mature fruit (FMF), which were harvested after floral induction at 18, 19, and 20 weeks, respectively, from the same commercial plantation, with 40 fruits allocated to each stage. The definition of the maturity stage followed the criteria established in our previous study [15]. To ensure robustness, three replicate samples were meticulously prepared for each biological sample, with 12 fruits sampled from each replicate to assess the physiological parameters. Fruit pulp weighing 20.0 g was collected following the methodology outlined in our previous work [42] and stored at −80 °C for subsequent assessments.

### 4.2. Evaluation of the Color Changes in Fruit

The color-related parameters, including L* (representing lightness), a* (indicating the transition from green to red), and b* (indicating the shift from blue to yellow) were measured using a high-precision colorimeter (HP-C210, Shanghai, China). To assess L*, a*, and b*, at least nine fruits at each ripening stage were halved longitudinally along the equator of the peel and pulp. At this juncture, three distinct points were selected for each parameter, and readings for L*, a*, and b* were recorded. For each pineapple, three readings were captured for each color-related parameter, ensuring robust and reliable measurements.

### 4.3. Measurements of Contents of Soluble Solids, Titratable Acid, and Vitamin C in Pulp

The total soluble solids content (TSS) was analyzed using a handheld PAL-1(B625333) device (ATAGO, Tokyo, Japan), and the values were expressed in degrees Brix.

For the titratable acid (TA) determination, we followed the acid–base titration method outlined by Li et al. [43]. Briefly, 3 g of frozen pulp powder was dissolved in 30 mL of distilled water and then heated to 80 °C for 30 min. After centrifugation at 4000× *g* for 15 min, 10 mL of the supernatant was titrated with 0.1 mol L^−1^ of NaOH, and the titration volume was recorded. The resulting data were expressed as the percentage of citric acid.

To determine the vitamin C (Vc) content, we employed a titration method utilizing 2,6-dichlorophenol indophenols following the procedure described by Hou et al. [44]. In short, 3 g of frozen pulp tissue was ground in 25 mL of 2% oxalic acid solution on ice, followed by centrifugation at 4000× *g* at 4 °C for 15 min. The supernatant was then titrated with 2,6-dichlorophenol indophenol, and the results were expressed as grams per kilogram (g kg^−1^) of fresh weight (FW).

### 4.4. Extractions and Assays of Contents of Total Phenolic, Total Flavonoids, and Lignin in Pulp

The total phenolic content was determined using the Folin–Ciocalteu method, as outlined by Zhou et al. [45], with slight modification. In brief, 0.5 g of frozen pulp tissue was homogenized in 80% (*v*/*v*) methanol for 2 h and subsequently centrifuged at 10,000× *g* for 15 min. The supernatant solution was collected for the analysis of total phenolic and flavonoid contents. A mixture consisting of 0.5 mL of the supernatant, 2 mL of Folin–Ciocalteu reagent, and 2 mL of 7.5% (*w*/*v*) Na_2_CO_3_ was incubated at 50 °C for 5 min. The absorbance was measured at 760 nm, and the results were expressed as grams of gallic acid equivalent per kilogram (g of gallic acid equivalent kg^−1^) of FW.

The total flavonoid content was determined using the aluminum nitrate method, as described by Zhou et al. [45]. In summary, 3 mL of supernatant, 0.5 mL of 5% (*w*/*v*) NaNO_2_, and 0.5 mL of 10% (*w*/*v*) AlNO_3_ were mixed. After 5 min, 1 mL of 1 mol L^−1^ NaOH was added. The absorbance was measured at 510 nm, and the results were calculated based on a standard curve and defined as g rutin kg^−1^ FW.

To determine lignin content, a modified version of the procedure described by Morrison [46] was employed. Approximately 5 g of pulp tissue powder was mixed with 95% pre-cooled ethanol and then centrifuged at 12,000× *g* for 30 min at 4 °C, and this process was repeated four times. Sediment was collected and dried overnight at 70 °C. Subsequently, 0.1 g of residue was dissolved in 1 mL of 25% (*v*/*v*) acetyl bromine–acetic acid, and the solution was incubated at 70 °C for 30 min. After cooling on ice, the reaction was terminated with 1 mL of 2 mol L^−1^ NaOH, followed by the addition of 2 mL of glacial acetic acid and 0.1 mL of 7.5 mol L^−1^ hydroxylamine hydrochloride (7.5 mL). After centrifugation at 12,000× *g* and 4 °C for 10 min, the absorbance was measured at 280 nm. The lignin content was then calculated against the standard curve and expressed as g.kg^−1^ of FW.

### 4.5. RNA Extraction, Illumine Sequencing, and Transcriptomics Data Analysis

Total RNA was meticulously extracted from three biological replicates of pulp tissues representing three distinct ripening stages, following the manufacturer’s instructions, using a Quick RNA isolation kit (Huayueyang, Beijing, China). RNA quality was comprehensively assessed using a 2100 Bioanalyzer RNA Nanochip (Agilent, Santa Clara, CA, USA). To confirm RNA integrity, electrophoresis was performed on formaldehyde-containing 1.5% (*w*/*v*) agarose gels.

For the library preparation, all procedures were performed in strict accordance with the manufacturer’s guidelines provided with the Truseq2 RNA sample prep Kit acquired from Illumina, Inc. San Diego, CA, USA. Raw data were acquired using the Illumina HiSeq TM2000 platform and subsequently aligned to the pineapple reference genome using HISAT2.

To identify differentially expressed genes (DEGs) in the samples, transcript abundance was estimated using the fragments per kilobase of exon per million mapped reads method. Statistically significant DEGs were identified based on the criteria of false discovery rate (FDR) ≤ 0.05, |log_2_ratio| ≥ 1. A *K*-means cluster analysis was carried out using the R package, and an enrichment analysis of the DEGs was performed using a hypergeometric distribution test. Gene expression patterns were visualized and presented using TBtools.

### 4.6. Weighted Gene Co-Expression Network Analysis and Gene Network Visualization

A total of 2194 DEGs were employed to construct unsigned co-expression networks using the weighted gene co-expression network analysis (WGCNA) tool, version 1.69 [47]. The following parameters were applied: a power of 14, a maximum module size of 5000, a minimum module size of 30, and a merge height of 0.25. For each module, the eigengene value was calculated and subsequently used to assess correlations with ripening properties including L*, a*, b*, total phenolic content, total flavonoid content, lignin content, TSS, Vc content, and TA. The co-expression diagram was visually represented using Cytoscape, version 3.8.1.

### 4.7. Quantitative Real-Time PCR (RT-qPCR) Verification

Total RNA isolation and cDNA synthesis were performed following a previously described method [42]. A total of fourteen DEGs, which were screened for RT-qPCR, with three biological replicates to verify expression levels, were involved in the biosynthesis of flavonoids and lignin. *Acactin* (HQ148720) was used as an endogenous reference gene, and gene-specific primer sequences are shown in Supplemental Appendix A. The relative expressions of these candidate genes were determined using the 2^−ΔΔCT^ method following the approach outlined by Livak and Schmittgen [48].

### 4.8. Statistics

Statistics analysis was performed using one-way analysis of variance (ANOVA) with SPSS (version 16.0, Chicago, IL, USA). The data were presented as the means ± standard error derived from three independent replicates. Statistically significant differences were assessed using the least significant difference test, with the significance set at *p <* 0.05.

## 5. Conclusions

The data presented in this study revealed significant changes in pineapple ripening. The metabolomic analyses demonstrated that an increased accumulation of flavonoids (ayanin, kaempferol 3-O-glucoside, chrysoeriol 5-O-hexoside, chrysoeriol O-hexosyl-O-hexosyl-O-glucuronic acid, acacetin O-acetyl hexoside, apigenin 7-O-glucoside, tricin O-malonylhexoside, tricin O-sinapoylhexoside, nobiletin, naringenin, and hesperidin), a reduction in lignin (sinapyl alcohol), and the degradation of organic acids (cis-aconitic acid, mandelic acid, rosmarinic acid, suberic acid, sebacate, and quinic acid O-di-glucuronic acid) might be attributed to yellow pigment, softening, and a decrease in acidity as the fruit ripened. Based on the integration of metabolomic and transcriptomic data, a series of putative candidate genes associated with the metabolisms of flavonoids, lignin, and organic acids including *FLS* (Aco027900), *FNS* (Aco019006), *F3H* (Aco001560 and Aco006882), *DFR*(Aco006769), *ANR*(Aco010710), *GST* (Aco001260, Aco013918, Aco005105, and Aco013915), *COMT* (Aco017593 and Aco018902), *CCR* (Aco015695), *CAD* (Aco001032), *POD* genes (Aco008465, Aco021646, Aco015271, Aco002056, Aco004613, Aco029136, Aco003045, Aco026779, Aco001617, Aco013666, Aco021355, Aco004784, and Aco021127), *LAC* (Aco015409), *APX* (Aco007028), *PPO* (Aco014848), and *ACO* (Aco009034) are inferred to play crucial roles in regulating the changes in fruit color and quality during pineapple ripening. The insights gained from this study will contribute to a better understanding of the molecular mechanisms underlying these ripening processes.

## Figures and Tables

**Figure 1 ijms-24-16384-f001:**
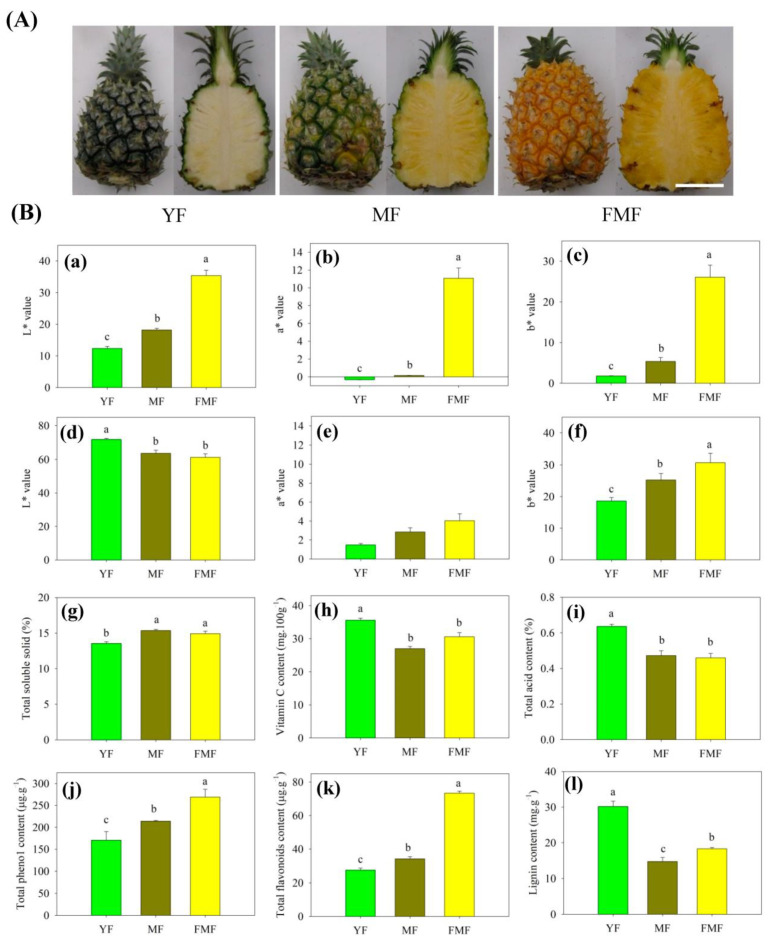
The changes in appearance (**A**) and (**B**) peel color B (**a**–**c**), pulp color B (**d**–**f**), Total soluble solid (Bg) (**g**), Vitamin C content (Bh) (**h**), Total acid content (Bi) (**i**), Total phenol content (Bj) (**j**), Total flavonoids content (Bk) (**k**), Lignin content (Bl) (**l**) of pineapple fruit at 18, 19, and 20 weeks after floral induction, respectively. Scale bar = 8 cm. Different letters indicate statistically significant differences (*p* ≤ 0.05).

**Figure 2 ijms-24-16384-f002:**
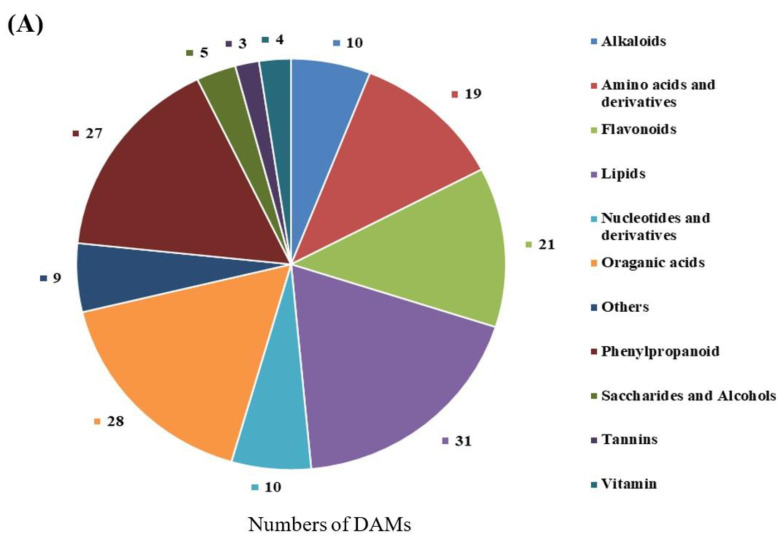
Preliminary analysis of 167 differentially accumulated metabolites (DAMs). (**A**) Pie chart of identified types and quantities of metabolites. (**B**) The red and blue bars indicate the numbers of upregulated and downregulated DAMs between YF and MF, YF and FMF, and MF and FMF, respectively.

**Figure 3 ijms-24-16384-f003:**
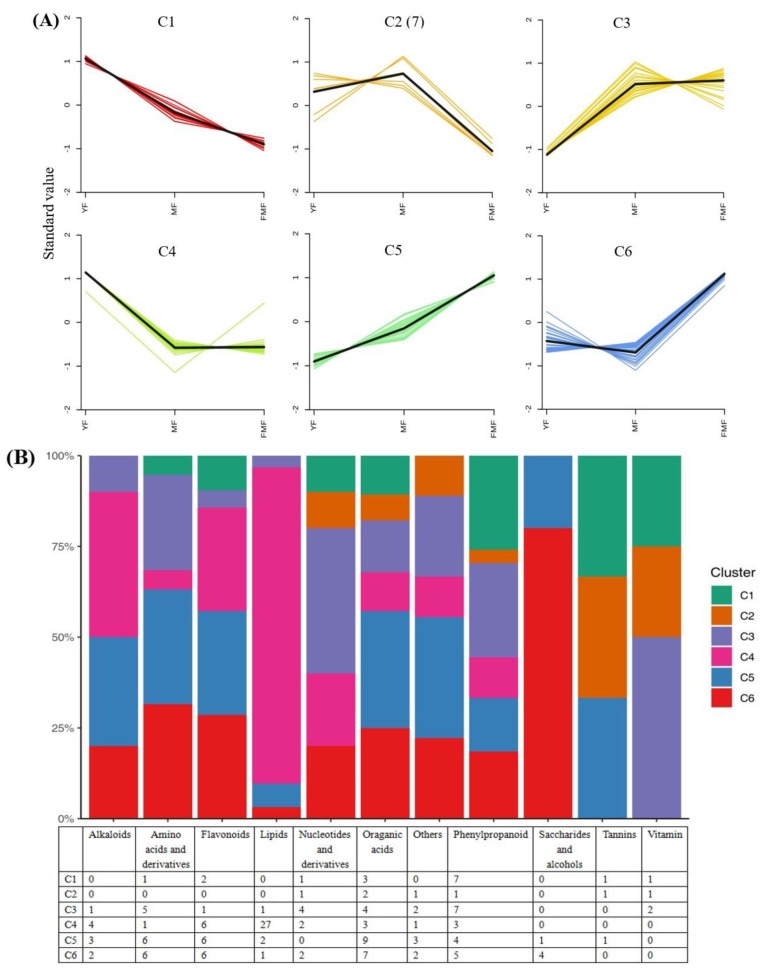
*K*-means clustering analysis of DAMs. (**A**) Six clusters of *K*-means (designated C1–C6). (**B**) The distribution of different substance categories in the 6 clusters. The table below the histogram shows the number of each type of metabolite in different clusters. (**C**) KEGG enrichment analysis for DAMs in the 6 clusters. (**D**) The top 10 pathways clustering analysis on the change trends of DAMs profiles in pineapple fruit from YF, MF, and FMF groups.

**Figure 4 ijms-24-16384-f004:**
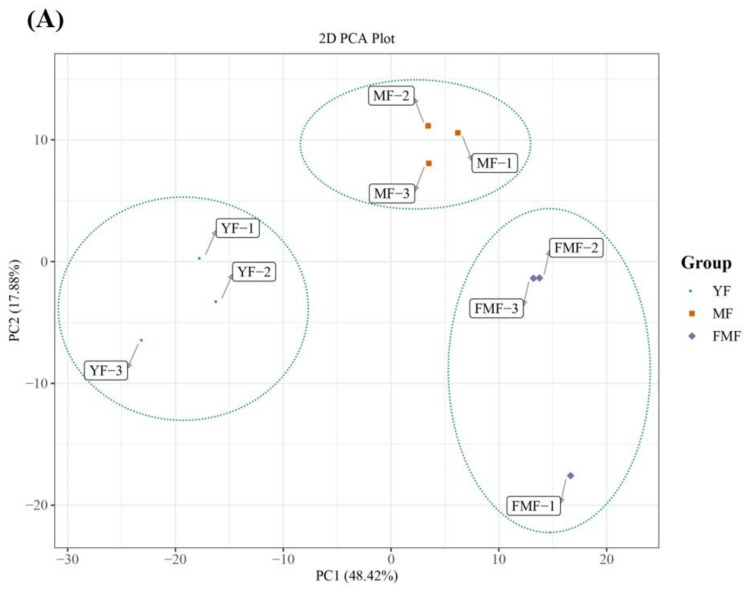
Preliminary analysis of 2094 differentially expressed genes (DEGs). (**A**) Principal component analysis (PCA) of the RNA-seq data. (**B**) Heatmap of 2094 DEGs from three developmental stages.

**Figure 5 ijms-24-16384-f005:**
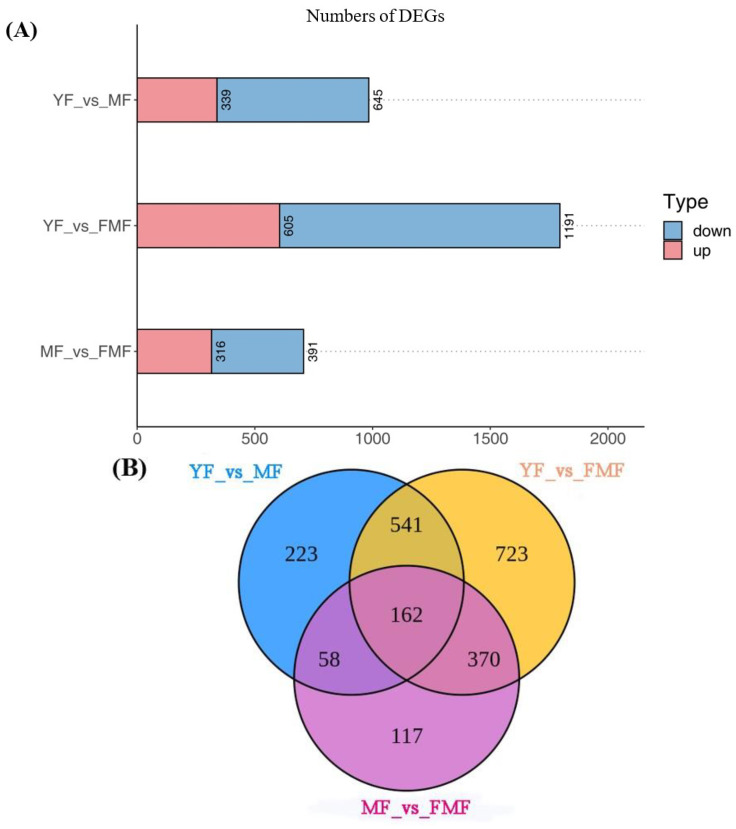
Preliminary analysis of 2194 differentially expressed genes (DEGs). (**A**) The numbers of upregulated and downregulated DEGs between YF and MF, YF and FMF, and MF and FMF, respectively. (**B**) Venn diagrams illustrating the overlap of DEGs revealed via paired comparison between YF and MF, YF and FMF, and MF and FMF, respectively. (**C**) Ten clusters of *K*-means (designated T1–T10). (**D**) KEGG enrichment analysis for DEGs in the 10 clusters.

**Figure 6 ijms-24-16384-f006:**
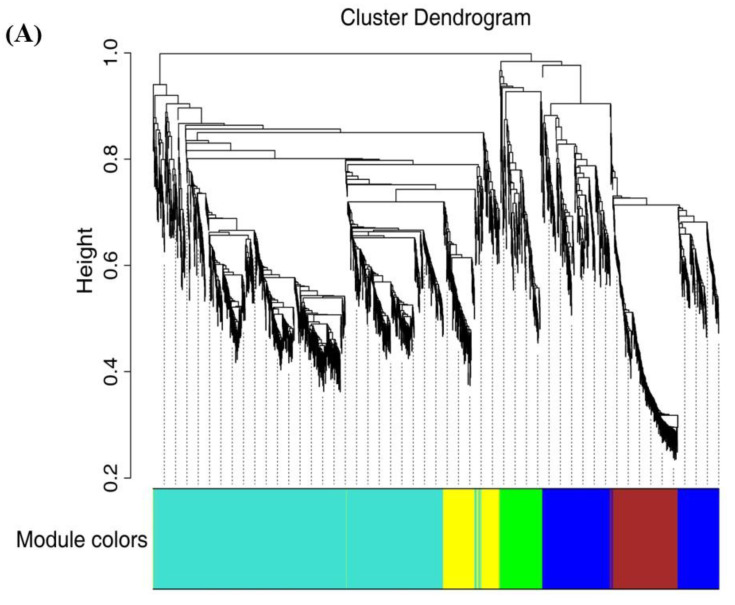
Correlations of 2194 DEGs with physiological indexes based on weighted gene co-expression network analysis (WGCNA). (**A**) Clustering dendrogram showing 6 modules of co-expressed genes with WGCNA. (**B**) Module–trait relationships. Each row corresponds to a module, and each column corresponds to a quality characteristic trait. Note: L*, L* value; a*, a* value; b*, b* value; TP, total phenols content; TF, total flavones content; TSS, total soluble solid; Vc, vitamin C; TA, total acid.

**Figure 7 ijms-24-16384-f007:**
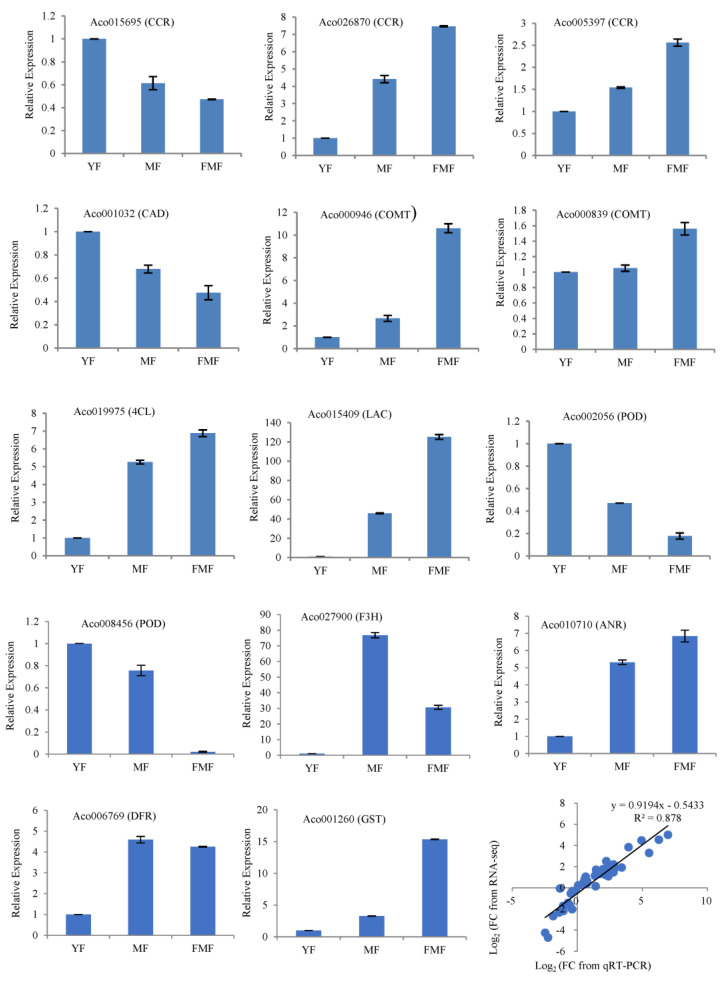
Expression levels of 14 candidate genes associated with the ripening of pineapple fruit via qRT-PCR and correlation analysis based on RNA-Seq and qRT-PCR data. Notes: FC, fold change.

**Figure 8 ijms-24-16384-f008:**
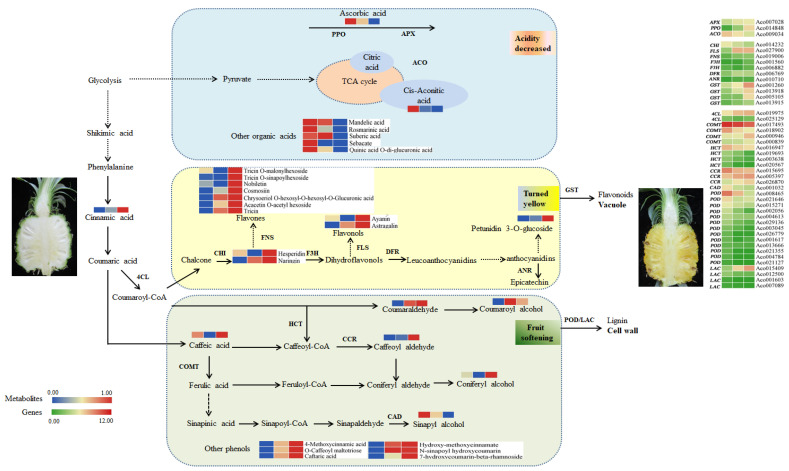
Dynamics of metabolites and genes related to pineapple fruit ripening. Note: Three blocks represent YF, MF, and FMF, respectively. SPS, sucrose phosphate synthase; APX, ascorbates peroxidase; ACO, aconitase; PPO, polyphenol oxidase; 4CL, p-coumarate: CoA ligase; CAD, cinnamyl alcohol dehydrogenase; CCR, cinnamoyl CoA reductase; COMT, caffeic acid O-methyl transferase; HCT, hydroxycinnamoyltransferase; POD, peroxidase; LAC, laccase; CHI, chalcone isomerase; F3H, flavanone 3-hydroxylase; DFR, dihydroflavonol-4-reductase; ANR, anthocyanidin reductase; LDOX, leucoantho-cyanidin dioxygenase; GST, glutathione transferases.

## Data Availability

The data presented in this study are available on request from the corresponding author.

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
