# Peer review of "Integration of Metabolomics and Transcriptomics to Explore Dynamic Alterations in Fruit Color and Quality in ‘Comte de Paris’ Pineapples during Ripening Processes"

_ijms, 2023, doi:10.3390/ijms242216384_

Round 1

Reviewer 1 Report

Comments and Suggestions for Authors

Dear authors, first of all, congratulations for the work realized. However, I have to understand some key data not included in the materials and methods. Please, read and answer the comments bellow:

I have a reasonable doubt about the methodology used to classify the diferents pineapple stages. I think that it is important to know if that fruit was harvested in different dates of the season or were used a maturity index (as in oranges) to take the decission as how to classify them. It is important for the robustness of your results to have a harvest pattern to repeat it again in the same conditions.

Was this assay repeated again? In other season? The environmental conditions of the field and the crop management could interfer in those results and conclusions. It is important to have data of differents seasons to check if those tendencies are right.

Why do not check the bioactive compounds of the peel? Because, the colour was measured in both, pulp and peel, but the following assays only were focused on pulp. Althought consumers only use the pulp. The marketing and shelf life is determined in high percentage by the peel. And maybe, the maturation on tree and ripening process after harvest in peel is different that in pulp.

Author Response

For research article

Response to Reviewer X Comments

1. Summary

Thank you very much for taking the time to review this manuscript. Please find the detailed responses below and the corresponding revisions/corrections highlighted/in track changes in the re-submitted files which is marked in red.

2. Questions for General Evaluation

Reviewer’s Evaluation

Response and Revisions

Does the introduction provide sufficient background and include all relevant references?

Yes/Can be improved/Must be improved/Not applicable

Are all the cited references relevant to the research?

Yes/Can be improved/Must be improved/Not applicable

Is the research design appropriate?

Yes/Can be improved/Must be improved/Not applicable

Are the methods adequately described?

Yes/Can be improved/Must be improved/Not applicable

Are the results clearly presented?

Yes/Can be improved/Must be improved/Not applicable

Are the conclusions supported by the results?

Yes/Can be improved/Must be improved/Not applicable

3. Point-by-point response to Comments and Suggestions for Authors

Comments 1: Dear authors, first of all, congratulations for the work realized. However, I have to understand some key data not included in the materials and methods. Please, read and answer the comments bellow:

Response 1: Thank you so much for your insightful comments and helpful suggestions. Those comments are valuable and helpful for revising and improving our manuscript. We have carefully modified the manuscript according to your suggestions. Following is the point-by-point responses.

Comments 2: I have a reasonable doubt about the methodology used to classify the diferents pineapple stages. I think that it is important to know if that fruit was harvested in different dates of the season or were used a maturity index (as in oranges) to take the decission as how to classify them. It is important for the robustness of your results to have a harvest pattern to repeat it again in the same conditions.

Response 2: Thank you so much for your insightful suggestion. The fruit was harvested in at 18, 19, and 20 weeks after floral induction from the same field in the same season.

Comments 3: Was this assay repeated again? In other season? The environmental conditions of the field and the crop management could interfer in those results and conclusions. It is important to have data of differents seasons to check if those tendencies are right.

Response 3: Thank you so much for your insightful suggestion. The assay was repeated in the same season from orchard with standardized management in Xuwen county of Zhanjiang city, Guangdong Province, China. As we know that the growth period of pineapple varied dependent on the season. But fruit harvested at 18, 19, and 20 weeks after floral induction are representative for three maturity stages.

Comments 4: Why do not check the bioactive compounds of the peel? Because, the colour was measured in both, pulp and peel, but the following assays only were focused on pulp. Althought consumers only use the pulp. The marketing and shelf life is determined in high percentage by the peel. And maybe, the maturation on tree and ripening process after harvest in peel is different that in pulp.

Response 4: Thank you so much for your insightful suggestion. In some cases, farmers usually sprayed ethylene on the peel of pineapple to make the peel yellow, but the quality of pulp remained unchanged. So the colour of peel doen’t represent the real ripening stages of pineapple. The edible quality is correlated closely with pulp, so we choose pulp as material.

4. Response to Comments on the Quality of English Language

5. Additional clarifications

Reviewer 2 Report

Comments and Suggestions for Authors

Interesting work about Metabolomics and Transcriptomics analysis of Fruit Color and Quality in Pineapple.

However, several revisions are required before publication:

Objectives of the work (lines 65-76) must be summarized without references only indicating the main purpose of the study.

Quality of Figures 4 and 6 must be improved increasing font size.

In Figure 7 the regression analysis must be clarified.

Discussion section is very week. Authors must clarify the novelty of the obtained results in comparison with previous pineapple fruit quality works. It is necessary to transform RNA-seq data in biological data, this is the nature of this manuscript. However, this biological data should be discussed in term of biological information adding some biological hypothesis clarifying the development of the fruit quality in pineapple. The election of some genes for the monitoring of this process is also very important in the Discussion section.

Description of the assayed pineapple cultivar must be clarified indicating (lines 71-73, page 23) min fruit quality characteristics and main phenological traits.

In the RNA-seq protocol biological and technical replications must be incorporated.

qPCR protocol must also be completed indicating technical and biological replications, reference genes and the selection of the assayed genes. It is also important to note if the RNA samples are different from the RNA-Seq analysis.

A new figure with the experimental design should be of great interest.

Conclusion section (lines 167-199) must be clarified indicating main implications of the obtained results from a production and breeding point of view.

Comments on the Quality of English Language

English grammar and expression should be revised.

Author Response

For research article

Response to Reviewer X Comments

1. Summary

Thank you very much for taking the time to review this manuscript. Please find the detailed responses below and the corresponding revisions/corrections highlighted/in track changes in the re-submitted files which is marked in red.

2. Questions for General Evaluation

Reviewer’s Evaluation

Response and Revisions

Does the introduction provide sufficient background and include all relevant references?

Yes/Can be improved/Must be improved/Not applicable

Are all the cited references relevant to the research?

Yes/Can be improved/Must be improved/Not applicable

Is the research design appropriate?

Yes/Can be improved/Must be improved/Not applicable

Are the methods adequately described?

Yes/Can be improved/Must be improved/Not applicable

Are the results clearly presented?

Yes/Can be improved/Must be improved/Not applicable

Are the conclusions supported by the results?

Yes/Can be improved/Must be improved/Not applicable

3. Point-by-point response to Comments and Suggestions for Authors

Comments 1: Objectives of the work (lines 65-76) must be summarized without references only indicating the main purpose of the study.

Response 1: Thank you so much for your constructive suggestion. We have resummarized as shown in line 63-74.

Comments 2: Quality of Figures 4 and 6 must be improved increasing font size.

Response 2: Thank you so much for your insightful suggestion. The font sizes of Figure 4 and 6 have been improved.

Comments 3: In Figure 7 the regression analysis must be clarified.

Response 3: Thank you so much for your insightful suggestion. The regression analysis  has been added as (A) analysis of expression level of key structural genes at different ripening stages of pineapple fruit; (B) correlation analysis based on RNA-Seq and qRT-PCR data. Notes: FC, fold change.

Comments 4: Discussion section is very week. Authors must clarify the novelty of the obtained results in comparison with previous pineapple fruit quality works. It is necessary to transform RNA-seq data in biological data, this is the nature of this manuscript. However, this biological data should be discussed in term of biological information adding some biological hypothesis clarifying the development of the fruit quality in pineapple. The election of some genes for the monitoring of this process is also very important in the Discussion section.

Response 4: Thank you so much for your insightful suggestion. Little modification has been made, which is marked in red, in the discussion section.

Comments 5: Description of the assayed pineapple cultivar must be clarified indicating (lines 71-73, page 23) min fruit quality characteristics and main phenological traits.

Response 5: Thank you so much for your insightful suggestion. ‘'Comte de Paris' pineapple, which is a popular cultivar in China for its abundant phenolic substances and easy management,’ was added in line 73 to explain the fruit quality characteristics and main phenological traits.

Comments 6: In the RNA-seq protocol biological and technical replications must be incorporated.

Response 6: Thank you so much for your insightful suggestion. ‘three biological replicates of’ and ‘with three technical replicates per biological relplicate’ were added in the RNA-seq protocol of the manuscript.

Comments 7: qPCR protocol must also be completed indicating technical and biological replications, reference genes and the selection of the assayed genes. It is also important to note if the RNA samples are different from the RNA-Seq analysis.

Response 7: Thank you so much for your insightful suggestion. I totally agreed with your opinion. Biological replications, reference genes and the selection of the assayed genes were indicated in the manuscript as shown in line 383-384, and ‘Notably, the RNA samples used were the same as the RNA in RNA-Seq analysis.’ was added in line 387 of the manuscript.

Comments 8: A new figure with the experimental design should be of great interest.

Response 8: Thank you so much for your insightful suggestion. The figure has been improved as shown in Fig.8.

Comments 9: Conclusion section (lines 167-199) must be clarified indicating main implications of the obtained results from a production and breeding point of view.

Response 9: Thank you so much for your insightful suggestion. ‘Our results can serve as a scientific foundation for molecular assisted breeding of pineapple for improving storage tolerance and fruit quality.’ was added in the conclusion section indicating the main implications of the obtained results from a production and breeding point of view.

4. Response to Comments on the Quality of English Language

Response: Thank you so much for your insightful suggestion. We have revised the manuscript which is marked in red through this manuscript.

5. Additional clarifications

For review article

Reviewer 3 Report

Comments and Suggestions for Authors

This manuscript is extensive, with studies including both metabolomics and transcriptomics investigating the ripening process of pineapple fruit. It is important that not only phytochemical compounds were identified but also genetic tests were carried out. It should be pointed out that the conducted genetic studies helped to identify putative genes associated with the metabolism during the ripening process. This is especially relevant for breeders.

The title corresponds to the content, the keywords are selected correctly.

Abstract. Need to be clarified because these findings are still preliminary or, as stated in the conclusions section, the role of genes is inferred. As the authors presented, further research is required to understand the regulatory mechanisms underlying ripening processes.

In the introduction, the authors analyse 15 literature sources (45 in the references list), briefly presenting the problem and what their research will reveal.

Results are presented in 7 figures. The information in the figures is clear enough to help readers understand the results even without the main text.

Discussion. The results are analysed in comparison with the works of other authors, the text is supplemented with Figure 8.

Materials and Methods. Plant material, amount and replications are described, and literature sources are indicated in the description of experimental methods, supplemented with modifications.

Conclusions. The conclusions section is particularly long, although it concentrates on the main results.

The references list is strangely arranged.

Minor remark. Unclear sentence (p.2, line 55).

Author Response

For research article

Response to Reviewer X Comments

1. Summary

Thank you so much for your comments and approval of our manuscript. These comments are valuable and helpful for revising and improving our manuscript. We have carefully taken the comments into consideration in preparing our revision. We submit here the revised manuscript as well as our point-by-point responses. All changed parts of the revised manuscript are marked in red. The following is a point-by-point response to the comments, and the responses are in blue.

.

2. Questions for General Evaluation

Reviewer’s Evaluation

Response and Revisions

Does the introduction provide sufficient background and include all relevant references?

Yes/Can be improved/Must be improved/Not applicable

Are all the cited references relevant to the research?

Yes/Can be improved/Must be improved/Not applicable

Is the research design appropriate?

Yes/Can be improved/Must be improved/Not applicable

Are the methods adequately described?

Yes/Can be improved/Must be improved/Not applicable

Are the results clearly presented?

Yes/Can be improved/Must be improved/Not applicable

Are the conclusions supported by the results?

Yes/Can be improved/Must be improved/Not applicable

3. Point-by-point response to Comments and Suggestions for Authors

Comments 1: The title corresponds to the content, the keywords are selected correctly.

Response 1: Thank you so much for your comments and approval.

Comments 2: Abstract. Need to be clarified because these findings are still preliminary or, as stated in the conclusions section, the role of genes is inferred. As the authors presented, further research is required to understand the regulatory mechanisms underlying ripening processes.

Response 2: Thank you so much for your insightful suggestion. I totally agree with your opinion and a more exact clarification was made in the abstract.

Comments 3: In the introduction, the authors analyse 15 literature sources (45 in the references list), briefly presenting the problem and what their research will reveal.

Response 3: Thank you so much for your comments and approval.

Comments 4: Results are presented in 7 figures. The information in the figures is clear enough to help readers understand the results even without the main text.

Response 4: Thank you so much for your comments and approval.

Comments 5: Discussion. The results are analysed in comparison with the works of other authors, the text is supplemented with Figure 8.

Response 5: Thank you so much for your comments and approval.

Comments 6: Materials and Methods. Plant material, amount and replications are described, and literature sources are indicated in the description of experimental methods, supplemented with modifications.

Response 6: Thank you so much for your comments and approval.

Comments 7: Conclusions. The conclusions section is particularly long, although it concentrates on the main results.

Response 7: Thank you so much for your comments and approval. We have made little modification and as shown in conclusion and the conclusion is shorter now.

Comments 8: The references list is strangely arranged.

Response 8: Thank you so much for your insightful suggestion. We made the format of references according to the requirements of the International Journal of Molecular Sciences journal.

Comments 9: Minor remark. Unclear sentence (p.2, line 55).

Response 9: Thank you so much for your insightful suggestion. Little modification has been made in line 55.

4. Response to Comments on the Quality of English Language

5. Additional clarifications

Round 2

Reviewer 2 Report

Comments and Suggestions for Authors

Authors have revised correctly the manuscript